



# Seasonal trends in the wintertime photochemical regime of the Uinta Basin, Utah, USA

Marc L. Mansfield, Seth N. Lyman

Department of Chemistry and Biochemistry, Utah State University Uintah Basin, Vernal, Utah 84078, USA

*Correspondence to*: Marc L. Mansfield (marc.mansfield@usu.edu)

**Abstract:**  Several lines of evidence indicate that the photochemical regime, i.e., the degree to which ozone production is either VOC- or $NO_x$-limited, varies with season in the Northern Hemisphere.  VOC-sensitivity seems to be more likely in winter and $NO_x$-sensitivity in summer.  For most regions, the question is patently academic, since excessive ozone occurs only in summer.  However, the Uinta Basin in Utah, USA exhibits ozone in excess of regulatory standards in both winter and

summer.  We have performed extensive F0AM box modelling to better understand these trends.  The models indicate that in late December the Basin's ozone system is VOC-sensitive, and either $NO_x$-insensitive or $NO_x$-saturated.  Sensitivity to $NO_x$ grows throughout the winter, and in early March, the system is about equally sensitive to VOC and $NO_x$.  The main driver for this trend is the increase in available solar energy as indicated by the noontime solar zenith angle.  A secondary driver is a decrease in precursor concentrations throughout the winter, which decrease because of, first, a dilution effect as

thermal inversions weaken, and second, an emission effect because certain emission sources are stronger at colder temperatures.  On the other hand, temperature and absolute humidity are not important direct drivers of the trend.





## 1. Introduction

The Uinta Basin of Utah and the Upper Green River Basin of Wyoming are the only locations worldwide with documented high wintertime concentrations of ozone that consistently exceed 70 ppb. This unique atmospheric phenomenon results because both basins are prone to persistent wintertime thermal inversions which trap ozone precursors in a tight boundary layer. A high surface albedo resulting from snow cover is also required (Schnell et al. 2009). Both basins are rural but are home to an active oil and natural gas extraction industry, which accounts for a major share of wintertime atmospheric emissions (Lyman et al. 2013 & 2018, Edwards et al. 2014). Interestingly, many urban valleys and basins have inversions and snow cover, but if anything, they are $NO_x$-saturated and titrate out ozone in winter (Shah et al. 2020, Li et al. 2021). The preferred explanation for the phenomenon is that the precursor speciation unique to the oil and gas industry is well-suited for winter ozone production (Schnell et al. 2009, Edwards et al. 2014, Ahmadov et al. 2015, Matichuk et al. 2017, Mansfield and Hall 2018).

Knowledge of the photochemical regime, or the degree to which an ozone system is either $NO_x$- or VOC-sensitive, is important in efforts to control ozone concentrations. It is well known that the ozone production efficiency, i.e., the number of ozone molecules generated for each $NO_x$ molecule consumed (defined operationally as the slope of the least-squares trend line of $O_x$ vs. $NO_z$ concentrations) indicates the relative photochemical regime, with larger values indicating a shift towards relatively higher $NO_x$-sensitivity and vice versa (Sillman 1995; Sillman et al., 1997; Sillman et al., 1998; Sillman, 1999; Rickard et al., 2002; Sillman and He 2002; Seinfeld and Pandis 2006; Chou et al., 2009). Another photochemical indicator with the advantage that it can be determined from satellite measurements is the ratio of the column densities of HCHO and $NO_2$. Larger values of the column $HCHO/NO_2$ ratio also indicate a shift towards $NO_x$-sensitivity (Martin et al., 2004; Duncan et al., 2010; Choi et al., 2012; Jin et al., 2017).

In many regions of North America, Europe, and East Asia, studies based on models or on measurements of photochemical indicators have observed seasonal trends in the photochemical regime. It is common to see ozone systems that are more $NO_x$-sensitive in summer and more VOC-sensitive in winter (Kleinman 1991; Jacob et al., 1995; Liang et al., 1998; Martin et al., 2004; Jin et al, 2017). In this paper, we report a similar trend in the Uinta Basin, Utah, USA. For most regions, the question of winter vs. summer ozone chemistry is purely academic, because ozone concentrations in exceedance of regulatory limits only occur in summer. However, in the Uinta Basin exceedances occur in winter and summer. Therefore, an understanding of the transition takes on added importance.


Below we report box model calculations to determine the drivers for this trend and to estimate sensitivities to $NO_x$ and VOC throughout the winter. The models indicate that in early winter, the Basin is either VOC-sensitive or $NO_x$-saturated, while in late winter, $NO_x$ and VOC sensitivities are about the same. We show that the main drivers for the trend are the change in solar zenith angle and a decrease in average precursor concentrations over the course of the winter. Other meteorological trends, specifically mean temperature and mean absolute humidity, are not important drivers. We also consider the factors driving the decrease in precursor concentrations. The data support a dilution effect as inversions become less intense during the advancing season. There is also evidence for an emission effect: Certain emission classes,





such as engine efficiency or equipment used more frequently in cold weather, are linked directly or indirectly to the temperature. This improved knowledge of the Basin's photochemical regime allows us to suggest possible ozone
abatement strategies.

Edwards et al. (2014) have also published box-model results for the Uinta Basin in winter. An important difference between their model and ours is the VOC speciation employed. Our speciation profile is reported below.

**2 Methodology**

**2.1 Photochemical Indicators in the Uinta Basin**

The Uinta Basin is a structural and sedimentary basin in eastern Utah, Fig. 1, that produces oil and natural gas. Unless stated otherwise, the data and models discussed here are from the Horsepool monitoring station at latitude 40.1434° and longitude –109.4689°. Figure 2 displays the ozone production efficiency, calculated daily as the slope of the least-squares
trend lines of $O_x$ vs. $NO_z$, and then averaged. Figure 3 displays the mean ratio of column HCHO data to tropospheric column $NO_2$ data obtained from the Ozone Monitoring Instrument (OMI, NASA-OMI 2022). The threshold between VOC and $NO_x$ sensitivity is near 1 or 2 for both indicators, but the precise threshold depends on local conditions, and it is best to interpret the indicators in light of modelling results, as we do below. Nevertheless, the data indicate a seasonal trend, with the system tending towards VOC sensitivity in early winter, and $NO_x$ sensitivity in late winter.


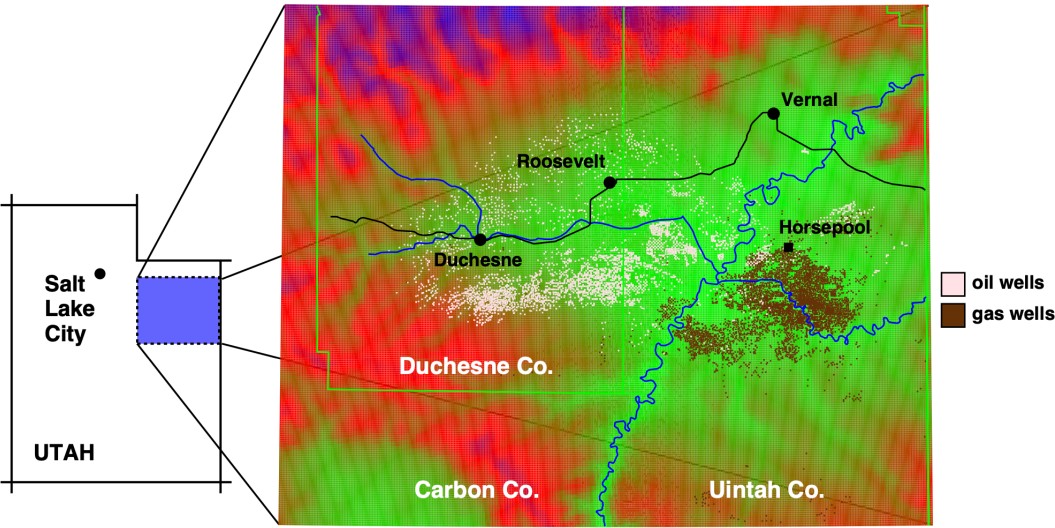

**Figure 1. Map of the Uinta Basin. Duchesne, Roosevelt, and Vernal are major population centers. The Horsepool monitoring station and the distribution of oil and natural gas wells are also shown. Background coloration indicates surface elevation.**





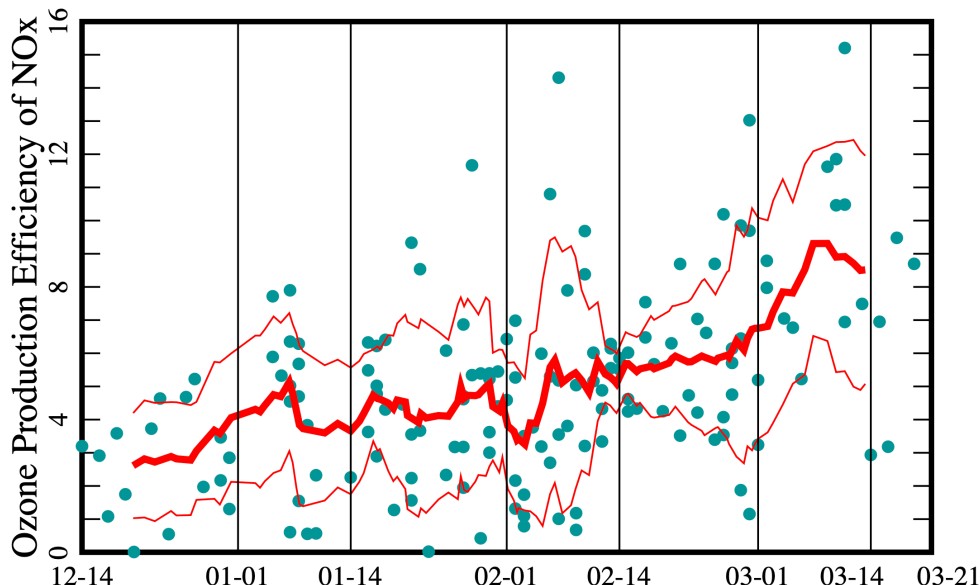


**Figure 2. Ozone production efficiency at the Horsepool monitoring station in the Uinta Basin. Data from days when the hourly ozone concentration exceeded 60 ppb from 2011 to 2022 and from December to March are shown. The red traces show ten-point running averages plus or minus one standard deviation.**

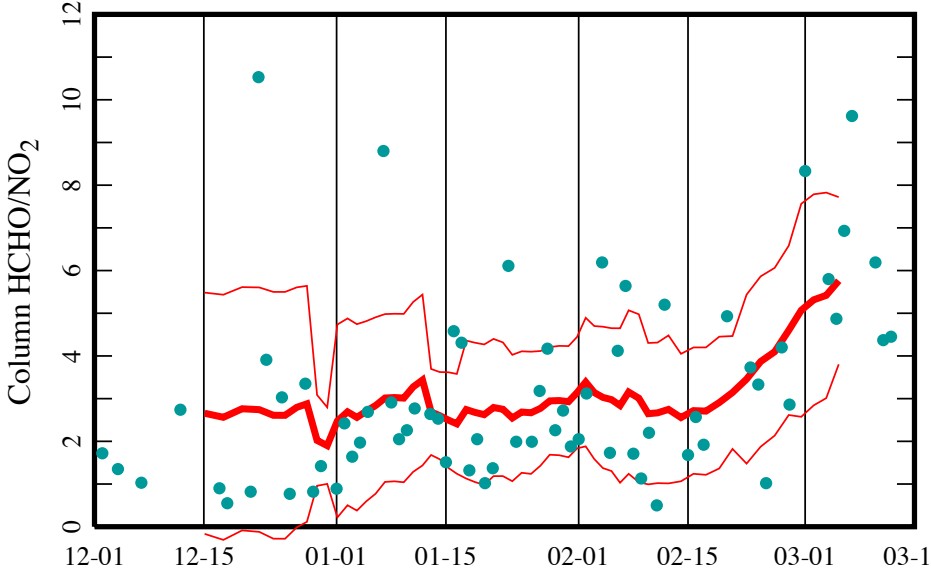


**Figure 3. Mean ratio of column HCHO to tropospheric column NO$_2$ from the Ozone Monitoring Instrument (OMI) pixel that contains the Horsepool station, on the indicated date, including data from 2009 to 2020 and between 1 December and 15 March. The red traces show 10-point running averages, plus or minus one standard deviation.**



## 2.2 Trends in meteorological variables and precursor concentrations

Any property that varies systematically through the season might conceivably be a driver for the trend in photochemical indicators seen in Figs. 2 and 3. This could include the actinic flux, the ambient absolute humidity, and the ambient temperature. Here, we take the noon-time solar zenith angle as a proxy for the actinic flux. This is permissible because during high ozone episodes, the sky is typically free of clouds and surface albedo contributed by snow cover is essentially uniform. The noon-time solar zenith angle is given to a good approximation by the formula

$$\theta = L + D \cos(\omega_E t) \tag{1}$$

where $L$ is the latitude (40.14° at the Horsepool station), $D$ is the tilt of Earth's axis (23.44°), $\omega_E$ is the angular frequency of Earth's revolution ($2\pi\ y^{-1}$), and $t$ is the time elapsed since the last winter solstice (Finlayson-Pitts and Pitts; 2000). Therefore, between the winter solstice and the vernal equinox, $\theta$ varies from about 63.6° to 40.1°. Figure 4 displays temperature and absolute humidity trends at the Horsepool station. Daily averages of temperature and absolute humidity between the hours of 11:00 to 20:00 MST, on dates from December 15 to March 15 and years between 2012 and 2021 are shown. (The rationale for computing means between the hours of 11:00 to 20:00 will be explained in Section 3, Calculation 4.) Throughout this work, the winter season has been divided into six fortnights or half-months, defined in Fig. 4. The fortnights will be designated "late December," "early January," and so on. Absolute humidity was calculated from measured values of temperature and relative humidity using a standard formula for the temperature dependence of the saturation vapor pressure of water (Seinfeld & Pandis; 2006).

$NO_x$ and methane concentrations are measured continuously at Horsepool during winter months, but non-methane organics are not. Therefore, we take methane as a marker for VOC concentrations, employing the conversion factor, explained below, of 0.0619 moles non-methane VOC for each mole of methane. Figure 5 indicates that $NO_x$ and methane concentrations are lower in late winter.

The available data indicate that a dilution effect caused by weakening inversions contributes to the systematic decrease in precursor concentrations. Tethersonde measurements do not occur on a regular basis in the Uinta Basin, so we rely on the correlation between surface temperature and altitude to obtain a quantitative measure of inversion strength (Mansfield and Hall, 2013; Mansfield and Hall, 2018). A similar approach has been adopted by other authors (Whiteman et al.; 2004; Largeron and Staquet; 2016). We define the daily "pseudo-lapse rate," $\Psi$, in terms of the slope of the least-squares trend line of the daily maximum surface temperature vs. altitude at a number of sites:

$$\Psi = -\frac{\partial T}{\partial z} \tag{2}$$

To exclude points that often lie in the non-linear region of the temperature-altitude profile, we only include sites between 1400 masl (the floor of the Basin) and 2000 masl (Mansfield and Hall, 2013; Mansfield and Hall, 2018). The daily maximum temperature is used to focus on persistent, as opposed to diurnal inversions. Low values of $\Psi$ indicate strong inversions with tight boundary layers, while high values indicate a well-mixed boundary layer. Figure 6 shows the variation in $\Psi$ as the season progresses. Inversions are seen to be more intense in early winter. Figure 7 shows the correlation between precursor concentrations





and Ψ, and Fig. 8 shows the correlation between $CH_4$ and $NO_x$ concentrations. These correlations all confirm that precursors are more diluted late in the season because the mixing layer is deeper.


Figure 4. **Absolute humidity and temperature trends throughout the winter. Humidity and temperature data are from the Horsepool monitoring station. Straight lines are the least-squares trend lines through the data points. Data are binned into six fortnights, late December, early January, etc. Black boxes show the 25th, 50th, and 75th percentiles. Red dots and whiskers show the mean plus or minus**
**one standard deviation.**



**Figure 5. Average daily NOx and CH₄ concentrations measured at Horsepool. Each symbol is a daily average taken over the hours 11:00 to 20:00 MST. Only days when both NO$_x$ and CH₄ data were reported have been displayed. Black boxes show the 25th, 50th, and 75th percentiles. Red dots and whiskers show the mean plus or minus one standard deviation.**





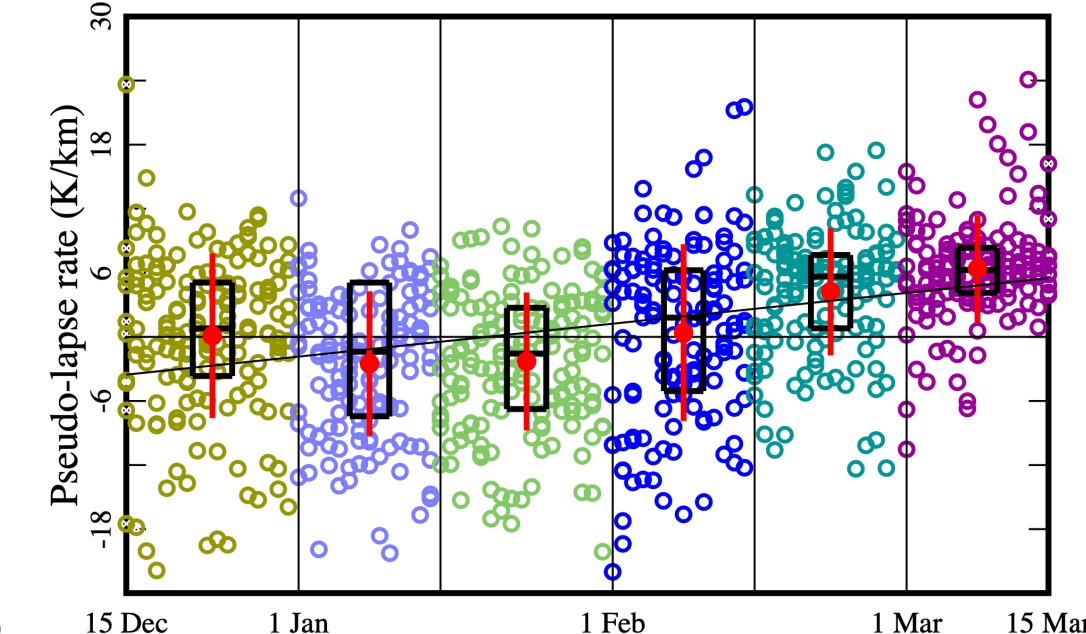


**Figure 6. Variation in pseudo-lapse rate Ψ as the season progresses. Each symbol represents the calculation for one day. Black boxes indicate the 25th, 50th, and 75th percentiles. Red dots and whiskers display the mean plus or minus one standard deviation.**





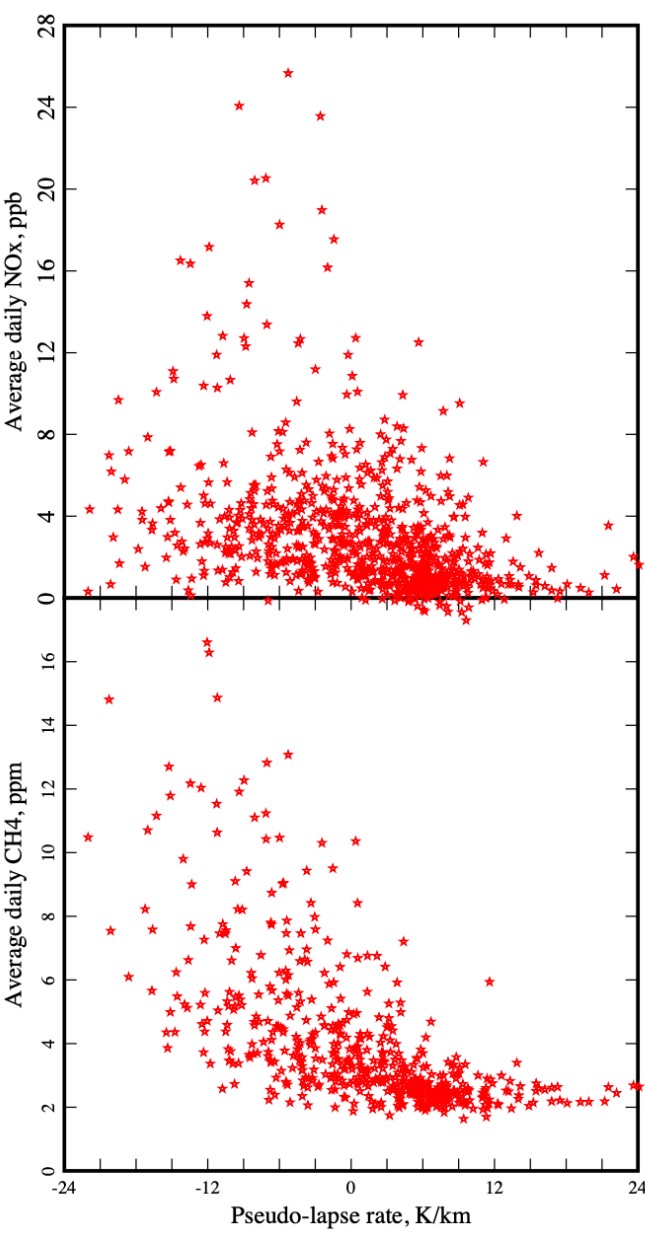

**Figure 7.  Correlations between NOₓ and methane concentrations and the pseudo-lapse rate Ψ.  Each symbol represents a daily average.**




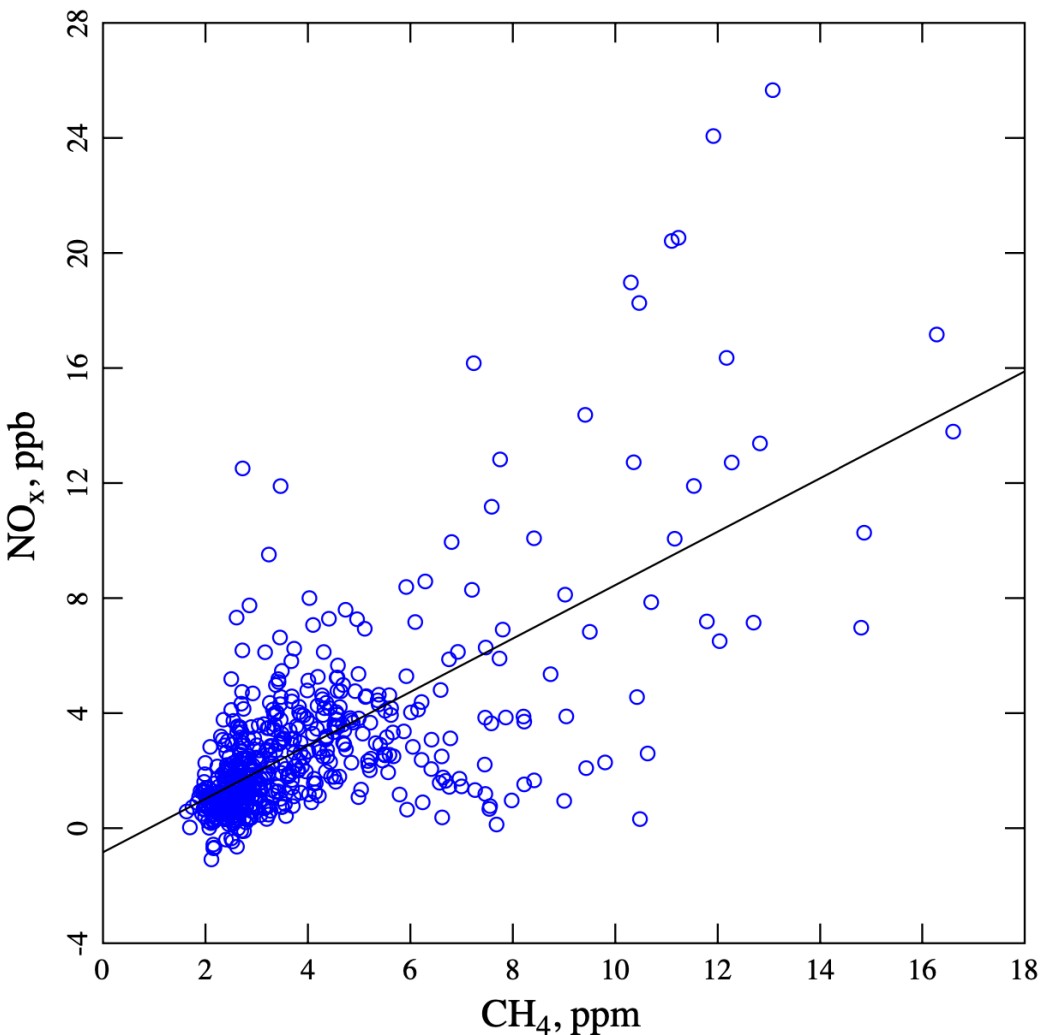

**Figure 8. Correlations between $NO_x$ and methane concentrations. Each symbol represents a pair of daily averages.**

However, emission effects might also contribute to the seasonal decline in precursor concentrations. Many studies find that vehicular $NO_x$ emissions are greater in winter. Several causative factors are mentioned, including poorer engine performance in the cold, cold starts, and operation of $NO_x$ after-treatment systems (e.g., catalytic converters) outside their optimal temperature range (Dardiotis et al., 2013; Reiter and Kockelman, 2016; Saha et al., 2018; Suarez-Bertoa and Astorga, 2018; Grange et al., 2019; Weber et al., 2019; Hall et al., 2020; Li et al., 2020; Wang et al., 2020; Bishop et al., 2022; Wærsted et al., 2022). Most studies

agree that the effect is present in light- and heavy-duty diesel vehicles (the predominant form of transportation near Horsepool), while studies investigating the effect in gasoline vehicles give conflicting results, perhaps because of differences in $NO_x$ after-treatment systems (Dardiotis et al.; 2013; Suarez-Bertoa and Astorga 2018; Grange et al., 2019; Li et al., 2020). Wærsted et al. (2022) report vehicular $NO_x$ emissions to be about a factor of three larger at –12 °C than at +12°C in Norway. Hall et al. (2020) report that vehicular $NO_x$ emissions in the Baltimore-Washington (USA) metropolitan area are about twice as large at –5 °C than



at 25 °C. Other studies give smaller ratios between summer and winter. Such variability likely results from variations in the composition of the local fleet (e.g., gasoline vs. diesel engines and older vs. newer $NO_x$ after-treatment systems). The Uinta Basin is also home to other $NO_x$ sources, such as well-site and portable natural gas-fueled heaters, that operate only in winter. We have been unable to find data on temperature trends in the $NO_x$ emissions from drilling rigs, but these may behave similarly to diesel-powered vehicles. Hence, it is likely that dilution and emission effects both contribute to the decrease in $NO_x$ concentrations. This

seasonal trend in $NO_x$ concentrations obviously deserves more study.

Figure 5 indicates that methane concentrations also vary systematically as the season progresses. Any dilution effects, cited above, will similarly affect methane concentrations. But emissions correlated with ambient temperature may also contribute, for example, from equipment such as glycol dehydrators, heat trace pumps, and "hot oil" trucks, and from operations to thaw frozen lines,

including pipeline venting and well blowdowns. Like $NO_x$, this methane trend also deserves further study.

Note that methane concentrations are rarely below 2 ppm, i.e., in late winter, they approach but never fall below the global background concentration (NOAA 2022). This is additional evidence of a well-mixed boundary layer under certain conditions. Also note that the ratio of the largest to the smallest mean in Fig. 4 is about 2 for methane and about 5 for $NO_x$. However, we

should not read too much into that difference. More relevant is the level of $CH_4$ relative to background.

**2.3 Box modelling procedures**

We used the "Framework for 0-D Atmospheric Modeling" (F0AM) platform, version 4.2.1, in a configuration similar to Lyman, et al. (2022), which has been coded as MATLAB script (Wolfe et al.; 2016). A subset of the "Master Chemical Mechanism,"

MCM v3.3.1, served as the chemistry mechanism (Jenkin et al., 2003; Saunders et al., 2003; Jenkin et al., 2015; Zong et al., 2018; MCM, 2022). Descriptions of all other input variables are summarized in Tables 1 and 2. Representative MATLAB code and input files are included as Supplementary Material. Each simulation spanned four days, including three days of spin-up. According to Table 2, one VOC unit is equivalent to 4920 ppb of methane and 304.4 ppb of the non-methane hydrocarbons, alcohols, and carbonyl compounds listed there.

Table 1. Inputs to the box models.

| Variable | Comments | Assigned value |
|---|---|---|
| VOC concentrations | Speciation profile from Table 2. One "unit" of input VOC implies the 59 compounds at the indicated concentrations. In any run, the total VOC concentration was set by scaling the total number of VOC "units." Species concentrations were held constant throughout the run (LinkSteps = 1, HoldMe = 1), equivalent to assuming VOC concentrations are at steady state with emission, deposition, and chemical transformation in balance. | One VOC unit = 4920 ppb $CH_4$ + 304.4 ppb of non-methane organics. |
| $NO_x$ concentrations | "Family conservation" option switched on; $NO_x$ = NO + $NO_2$ concentrations held constant throughout each hour. Hourly NO and $NO_2$ profiles prepared in several ways: (1) Observational data from a single day. (2) Averages of observational data from a number of | |



| | days. (3) Rescaling any of the profiles prepared by the previous two ways. | |
|---|---|---|
| Background ozone | Lyman et al., 2013; Lyman et al. 2018. | 50 ppb |
| Temperature | Hourly data from a given day. | |
| Relative humidity | Hourly data from a given day. | |
| Barometric pressure | Hourly data from a given day. | Average over all models: 845 ± 7 (1σ) mbar. |
| CO concentration | Hourly concentration measurements from Horsepool on 2019-02-27.[a] CO concentration held constant throughout each hour (LinkSteps = 1, HoldMe = 1). | 272 ppb (average from 11:00 to 20:00 MST) |
| $k_{dil}$, dilution factor | Same as Edwards et al., 2014. | $1.8 \times 10^{-5}\ hz$; daylight; $2.0 \times 10^{-6}\ hz$; dark |
| Solar zenith angle | Computed with the F0AM procedure sun_position from the time, date, latitude, longitude, and elevation at Horsepool. | |
| Albedo | Typical of snow surfaces at Horsepool. | 0.7 |
| Ozone column | From Giovanni NASA website (NASA, 2022). | 275 DU |
| $J_{corr}$ | Correction factor for scaling J-values (Wolfe et al., 2016). | 0.5 |

[a]Due to an oversight, CO concentration data from 2019-02-27 were applied in all modeling runs. However, we verified that modeled ozone concentrations changed by no more than about 1 ppb even when we completely zeroed out the CO concentration.


Table 2. VOC concentrations in the F0AM box model (Lyman et al.; 2021). The concentrations listed here constitute one "unit" of VOC in the models.

| Methane | | Total | Aromatics | | Total |
|---|---|---|---|---|---|
| methane | 4920.0 ppb | 4920.0 ppb | benzene | 1.2 ppb | 3.0 ppb |
| Non-methane Alkanes | | Total | toluene | 1.2 | |
| ethane | 123.0 ppb | 264.8 ppb | o-xylene | 0.1 | |
| propane | 63.0 | | m-xylene | 0.2 | |
| n-butane | 25.0 | | p-xylene | 0.1 | |
| Isobutane | 15.0 | | ethylbenzene | 0.1 | |
| n-pentane | 10.0 | | 1,2,3-trimethylbenzene | 0.1 | |
| Isopentane | 11.0 | | n-propyl benzene | 0 | |
| n-hexane | 4.0 | | isopropyl benzene | 0 | |
| 2-methylpentane | 3.0 | | 1,2,4-trimethylbenzene | 0 | |





| 3-methylpentane | 2.0 | | 1,3,5-trimethylbenzene | 0 | |
| 2,2-dimethylbutane | 0.3 | | 1-ethyl-2-methylbenzene | 0 | |
| 2,3-dimethylbutane | 1.6 | | 1-ethyl-3-methylbenzene | 0 | |
| n-heptane | 1.9 | | 1-ethyl-4-methylbenzene | 0 | |
| 2-methylhexane | 0.8 | | styrene | 0 | |
| 3-methylhexane | 1.2 | | Alcohols | | Total |
| n-octane | 0.8 | | methanol | 10.0 ppb | 12.6 ppb |
| n-nonane | 0.2 | | ethanol | 0.3 | |
| n-decane | 0.2 | | isopropanol | 2.3 | |
| cyclohexane | 1.8 | | Carbonyls | | Total |
| Alkenes and Alkynes | | Total | formaldehyde | 6.5 ppb | 21.0 ppb |
| ethylene | 1.1 ppb | 3.0 ppb | acetaldehyde | 2.9 | |
| propylene | 0.1 | | butyraldehyde | 1.3 | |
| acetylene | 1.8 | | acrolein | 1.6 | |
| 1-butene | 0 | | methacrolein | 0.6 | |
| cis-2-butene | 0 | | benzaldehyde | 4.4 | |
| trans-2-butene | 0 | | acetone | 3.0 | |
| 1-pentene | 0 | | methyl ethyl ketone | 0.7 | |
| cis-2-pentene | 0 | | propionaldehyde | 0 | |
| trans-2-pentene | 0 | | valeraldehyde | 0 | |
| 1-hexene | 0 | | crotonaldehyde | 0 | |
| | | | cyclohexanone | 0 | |


Often, we wanted to compare two models with the same absolute humidity profile but at different temperatures. To change the temperature without changing the absolute humidity of course requires an adjustment of the relative humidity.


Typically, ozone concentrations in the Basin grow for several days during a multi-day inversion episode, peaking near the end of the episode. [Lyman et al. 2013 & 2018] The models described below examined peak-ozone days.

**2.4 Definition of sensitivity**

Let $x$ represent some independent variable in the model, for example, the concentration of a precursor, and let $y$ represent a dependent variable, which for our purposes is almost always the maximum ozone concentration on day four of a simulation run. A small variation in the independent variable, $dx$, induces a change, $dy$, in the dependent variable. The fractional changes in the two variables are $dx/x$ and $dy/y$. We define the sensitivity, $S$, of the dependent variable on the independent variable as the ratio of

these two fractional changes:





$$S = \left(\frac{dy}{y}\right)/\left(\frac{dx}{x}\right) = \frac{x}{y}\frac{dy}{dx} = \frac{d\ln y}{d\ln x} \tag{3}$$

A small, 1% say, change in $x$ produces an $S\%$ change in $y$. By this definition $S$ is unitless and is the slope of the tangent on a log-log plot. When the independent variable is the $NO_x$ or VOC concentration, we will use the notation $S_{NOx}$ and $S_{VOC}$, respectively. $S$ values were calculated by numerical differentiation using three separate runs at $x$ and $x \pm dx$ for $dx$ on the order of a few percent. We use the phrases "NOx-saturated" to indicate $S_{VOC} > 0 > S_{NOx}$, "VOC-sensitive" to indicate $S_{VOC} > S_{NOx} > 0$, and "NOx-sensitive" to indicate $S_{NOx} > S_{VOC} > 0$.

## 3 Results

We performed five separate calculations to probe the effect of the systematic variations documented in Section 2.2.

### 3.1 Calculation 1. $NO_x$ and VOC sensitivity of 24 different models.

We identified 24 peak-ozone days between 15 December and 15 March and between 2013 and 2021. Maximum one-hour ozone concentrations on these days varied anywhere from 59 to 154 ppb. Observational values of meteorological data and of $NO_x$ concentrations were employed as input data. Input VOC concentrations were in the proportion given in Table 2. For each model, we defined a "base case" using the observed $NO_x$ concentrations and by scaling the input VOC concentration until the maximum day-4 ozone concentration agreed with measurements. These 24 models are summarized in Table 3. Figure 9 shows how the VOC and $NO_x$ sensitivities of each of the base-case models vary throughout the season. In December and January, VOC sensitivities are always larger than $NO_x$ sensitivities, and $NO_x$ sensitivities are often negative. In late winter, $NO_x$ and VOC sensitivities are typically comparable. The three late-winter base-case models (2019-02-27, 2013-03-03, 2019-03-06) have nearly equal sensitivities to $NO_x$ and VOC.

Table 3. List of F0AM models constructed.

| Model | Date | Base-case concentrations | | Mean temperature, Celsius | Mean absolute humidity, mbar |
|---|---|---|---|---|---|
| | | VOC (multiples of Table 1) | $NO_x$, ppb (average between 11:00 and 20:00 MST) | | |
| D13a | 2013-12-16 | 2.25 | 7.04 | −12.53 | 1.75 |
| D20a | 2020-12-21 | 0.30 | 4.91 | −6.50 | 3.19 |
| D13b | 2013-12-30 | 1.6 | 6.24 | −10.45 | 2.21 |
| J21a | 2021-01-05 | 0.5 | 3.41 | −5.34 | 2.99 |
| J15a | 2015-01-07 | 0.9 | 13.07 | +1.76 | 5.17 |
| J20a | 2020-01-08 | 0.45 | 3.32 | −8.28 | 2.78 |
| J13a | 2013-01-10 | 1.7 | 12.05 | −13.15 | 1.77 |
| J16a | 2016-01-17 | 0.65 | 2.83 | +0.88 | 4.05 |
| J21b | 2021-01-17 | 0.5 | 2.81 | −4.64 | 3.01 |



| J13b | 2013-01-26 | 2.5  | 4.62 | −3.31 | 4.06 |
| J14a | 2014-01-27 | 0.75 | 2.47 | −1.48 | 3.72 |
| J16b | 2016-01-29 | 0.65 | 3.51 | −4.48 | 3.31 |
| F19a | 2019-02-01 | 0.60 | 2.74 | −6.84 | 2.52 |
| F13a | 2013-02-06 | 1.3  | 4.65 | −3.50 | 3.53 |
| F17a | 2017-02-06 | 4.00 | 0.76 | +1.71 | 5.05 |
| F20a | 2020-02-06 | 0.25 | 2.73 | −6.56 | 2.48 |
| F14a | 2014-02-08 | 0.5  | 4.56 | +2.77 | 4.69 |
| F16a | 2016-02-12 | 1.00 | 3.18 | +0.21 | 4.44 |
| F19b | 2019-02-14 | 0.55 | 2.87 | −2.72 | 4.53 |
| F13b | 2013-02-17 | 2.4  | 1.81 | −6.21 | 2.95 |
| F13c | 2013-02-21 | 1.05 | 3.14 | −2.74 | 3.68 |
| F19c | 2019-02-27 | 0.85 | 1.35 | −3.24 | 3.63 |
| M13a | 2013-03-03 | 1.55 | 2.29 | +1.30 | 4.98 |
| M19a | 2019-03-06 | 0.15 | 1.12 | −1.52 | 5.13 |

The late-winter convergence of $S_{VOC}$ and $S_{NOx}$ appears to result more from an increase in $S_{NOx}$ than from a decrease in $S_{VOC}$. The trend line for $S_{VOC}$ has a slope of −1.6 σ, for σ the standard deviation of the slope, and a *p*-value of 0.13 from a Mann-Kendall trend test (Mann, 1945; Kendall, 1975). Therefore, the downward trend in $S_{VOC}$ may be real, but our results lack sufficient statistical power to confirm this. On the other hand, with a slope of +3.9 σ and a very small Mann-Kendall *p*, the upward trend in $S_{NOx}$ is statistically significant.



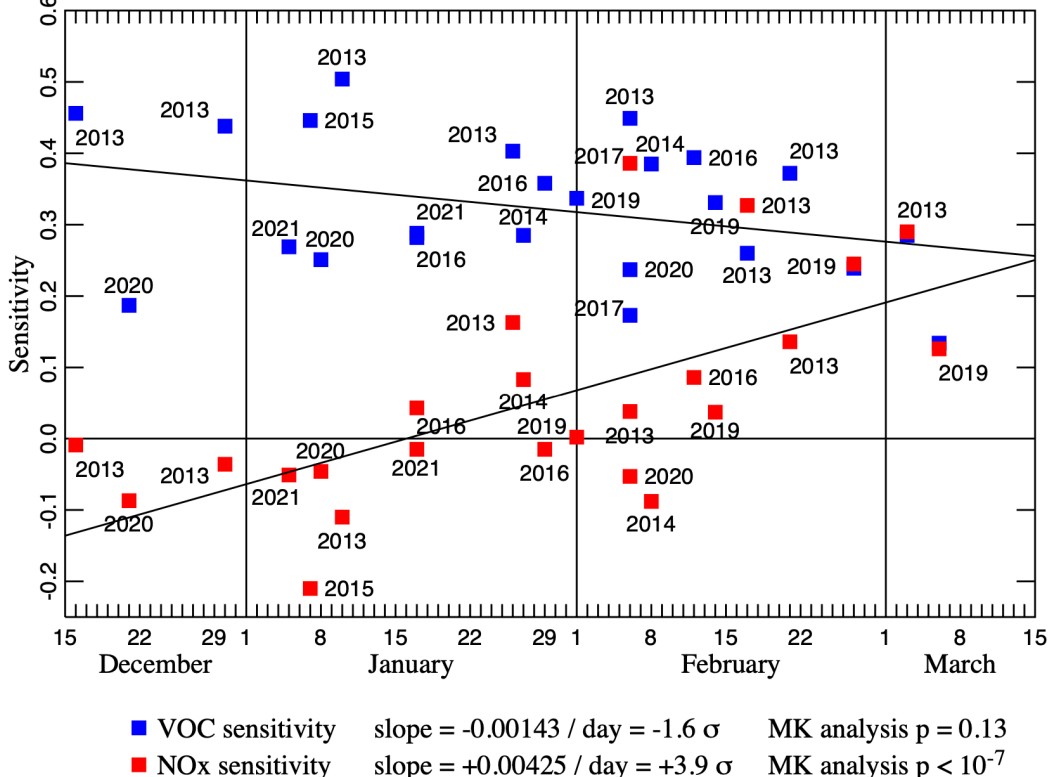

| | slope = -0.00143 / day = -1.6 σ | MK analysis p = 0.13 |
| VOC sensitivity | | |
| NOx sensitivity | slope = +0.00425 / day = +3.9 σ | MK analysis p < 10⁻⁷ |

**Figure 9.** $S_{VOC}$ and $S_{NOx}$ for 24 different box model runs. Slopes of the least-squares trend lines are given both as numerical values and as multiples of the standard deviations of the slopes. The *p*-values from Mann-Kendall trend analyses are also shown.

### 3.2 Calculation 2. Impact of changing meteorological and concentration variables.

We examined five variables as candidate drivers for the change in photochemical regime documented above, namely, VOC concentration, $NO_x$ concentration, solar zenith angle (SZA), ambient temperature (T), and ambient absolute humidity (AH). Starting from one of the models in Table 3, we replaced just one of the above five variables with typical values from either an earlier or a later model. No other input quantity was varied. The solar zenith angle was varied by changing the date of the simulation. This lets us probe the effect of just one variable on $S_{NOx}$ and $S_{VOC}$. Figure 10 plots the change in sensitivity, $\Delta S_{NOx}$ or $\Delta S_{VOC}$, induced by the change in one of the five variables, solar zenith angle (Δ SZA), $NO_x$ concentration (Δ $NO_x$), VOC concentration (Δ VOC), temperature (Δ T), and absolute humidity (Δ AH). For ease of interpretation, whenever one of the five variables trends downward during the winter (Equation 1 and Figs. 4 and 5) the scale of the corresponding abscissa in Fig. 10 is shown in descending order. In this way, we see immediately whether modulating the variable tends to increase or decrease the sensitivity over the course of the winter. The scale of the ordinates in either row is identical, so the vertical displacement of each trend line indicates the relative strength of each individual driver. Most notably, the response to AH is very weak, indicating that absolute humidity is not an important driver. Traditionally, either the condition |slope|/σ > 2 or $p_{MK} < 0.05$ indicates a statistically significant trend in a dataset. Therefore, of the eight remaining trends depicted in Fig. 10, the $\Delta S_{VOC}$ vs. Δ $NO_x$ is least likely to be statistically significant. Normally, we expect each trend line to pass near the origin: A small change in only one variable usually





leaves the model largely unchanged. The two trend lines relative to $\Delta NO_x$ fail to pass near the origin because the $NO_x$ variable is defined as an average over a daily profile. Therefore, $\Delta NO_x = 0$ can occur even if the models are not equivalent. This also explains the greater scatter in the two $\Delta NO_x$ plots.

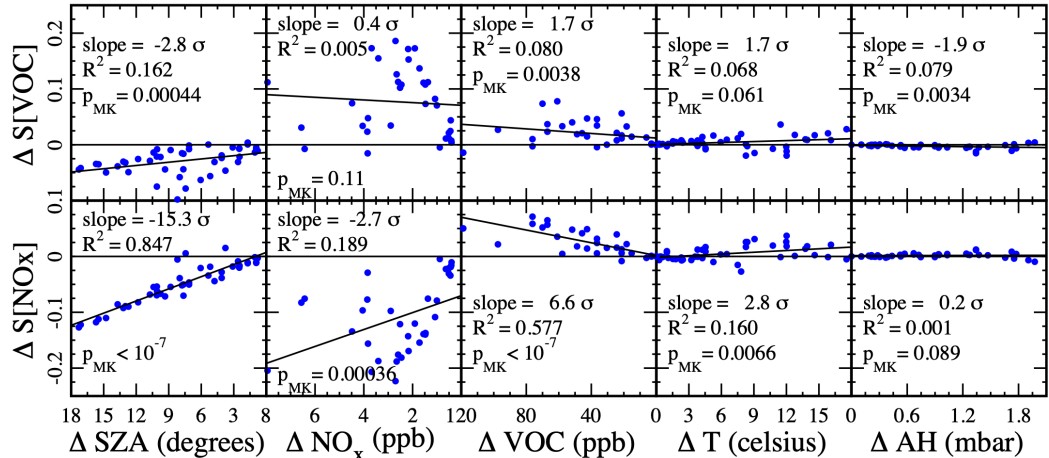

**Figure 10. The change in sensitivity, $\Delta S_{NOx}$ or $\Delta S_{VOC}$, induced by the change in one of the five variables, solar zenith angle ($\Delta$ SZA), $NO_x$ concentration ($\Delta$ NOx), VOC concentration ($\Delta$ VOC), temperature ($\Delta$ T), and absolute humidity ($\Delta$ AH) for various models listed in Table 3. The scale on the abscissa appears in descending order if the corresponding variable trends downward during the winter. The slope of each trend line as a multiple of its standard deviation, the value of Pearson's $R^2$, and the Mann-Kendall *p*-value are shown for each data set.**

The total vertical displacement of each of the trend lines in Fig. 10 is one measure of the contribution of each of the five drivers to the seasonal sensitivity trends. Trend lines in the temperature and absolute humidity plots are relatively flat, indicating that the solar zenith angle and the $NO_x$ and VOC concentrations are the important drivers. Table 4 displays the values of all vertical displacements. Uncertainties were calculated from the standard deviations of the slopes. Sums of all five displacements are also tabulated. The uncertainty in the sums was calculated by propagating the uncertainties in each addend into the sum. The $\Delta S_{VOC}$ total is statistically zero, consistent with the finding that the $S_{VOC}$ trend in Figure 9 may not be statistically significant. The solar zenith angle and the NOx concentration make comparable positive contributions to $S_{NOx}$, while the VOC concentration makes a smaller negative contribution.

Table 4. Vertical displacement of trend lines in Figure 10.

| Variable | $\Delta S_{VOC}$ | $\Delta S_{NOx}$ |
|---|---|---|
| SZA | $0.035 \pm 0.012$ | $0.131 \pm 0.009$ |
| NOx | $-0.019 \pm 0.046$ | $0.121 \pm 0.044$ |
| VOC | $-0.025 \pm 0.015$ | $-0.068 \pm 0.010$ |
| T | $0.010 \pm 0.006$ | $0.019 \pm 0.007$ |
| AH | $-0.004 \pm 0.002$ | $0.0005 \pm 0.0022$ |
| TOTAL | $-0.003 \pm 0.050$ | $0.20 \pm 0.05$ |





### 3.3 Calculation 3. Ozone isopleth diagrams.

We calculated an ozone isopleth diagram for each of the 24 models in Table 3 by scaling $NO_x$ and VOC concentrations relative to the base model. Each F0AM run required approximately 2 to 4 minutes on a MacBook Pro laptop and generating the full diagram at high resolution proved to be too time-consuming. Rather, we calculated pixels at high resolution only around the boundary of the diagram, and at lower resolution throughout the interior. Pixels were also calculated at higher resolution in the vicinity of the "indicator curves," to be defined below. The ozone isopleth surface at all remaining pixels was generating by kriging interpolation

(Kerry and Hawick, 1998). All 24 diagrams are given in the Supplementary Material.

### 3.4 Calculation 4. Superposition of individual models to create ozone isopleth diagrams for each fortnight.

According to Calculation 2, the only relevant variables are solar zenith angle and the precursor concentrations. This implies that all isopleth surfaces belonging to any one fortnight are approximately superposable. The VOC concentration unit, defined in reference to Table 2, is directly transferable between different models, but the $NO_x$ concentration unit, defined in reference to the

daily $NO_x$ profile, is not. To test superposability, we assigned a different scale factor to the $NO_x$ axis of each diagram and adjusted its value to optimize the superposition among all models from a given fortnight. We found that the scale factor for each model correlated best with the average $NO_x$ concentration taken over the hours 11:00 to 20:00 MST and adopted this average to redefine the $NO_x$ concentration axis on the ozone isopleth surfaces. For consistency, we have also used averages over the same hours, 11:00 to 20:00 MST, to report values of other variables.


Ozone isopleth surfaces for each of the 24 models listed in Table 3 were then superposed into six composite surfaces, one for each fortnight, and are shown in Fig. 11. All contributions defined at a given pixel on the original surfaces were averaged to estimate the ozone concentration in the composite pixel. Discontinuities in the contour curves occur because the diagrams of the individual models were not constructed with the same boundaries and because the individual models are not perfectly superposable.

Nevertheless, the discontinuities are generally not large, validating the superposability assumption. The domains defined by the 25[th] and 75[th] percentiles in Figs. 4 and 5 appear in Fig. 11 as white rectangles, and therefore define the domains at which precursor concentrations are at their typical values. The small white squares indicate points at which $S_{NOx} = 0$. The small pink squares indicate points at which $S_{NOx} = S_{VOC}$. These small squares define indicator curves: All points below the pink trace satisfy $S_{NOx} > S_{VOC}$ and constitute the region of $NO_x$ sensitivity. All points above the white trace satisfy $S_{NOx} < 0$ and constitute the $NO_x$-saturation

region. All points between the two traces satisfy $S_{VOC} > S_{NOx} > 0$ and constitute the VOC-sensitive region. The indicator curves from each individual model are shown. The fact that these are all approximately superposable is further vindication of the superposability assumption.



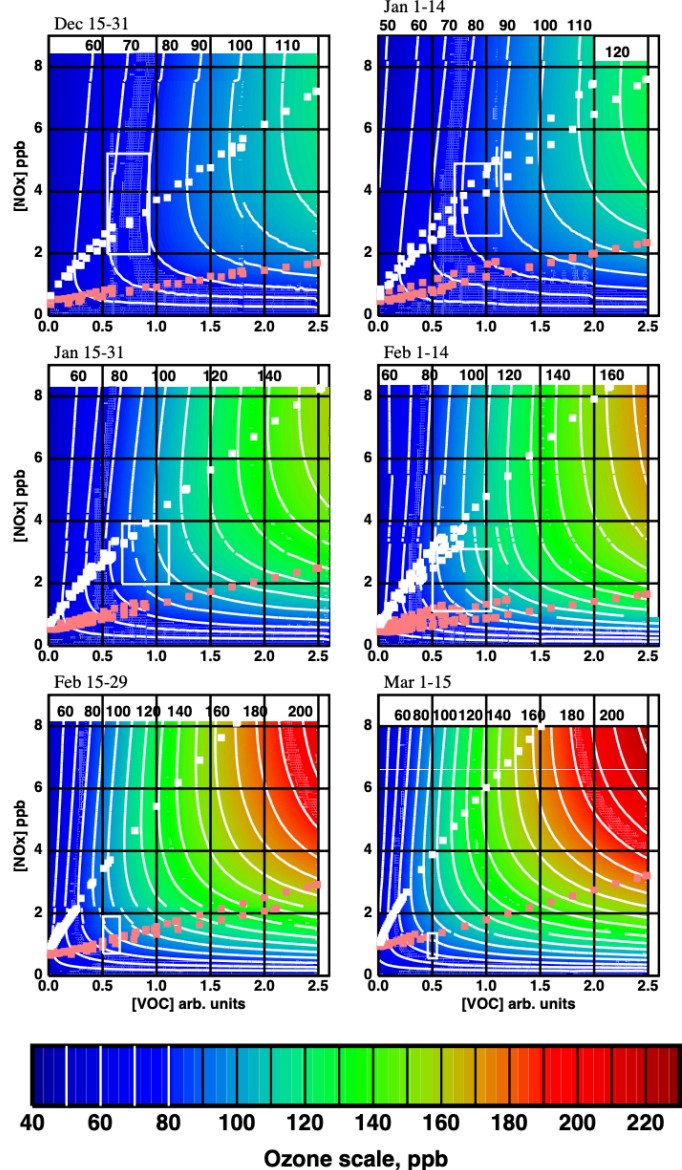


**Figure 11. Composite ozone isopleth surfaces constructed from the 24 models listed in Table 3. White squares define the locus of points at which $S_{NOx} = 0$. Pink squares define the locus of points at which $S_{NOx} = S_{VOC}$. White rectangles were derived from the 25th-to-75th percentile boxes in Figs. 4 and 5 and therefore display the typical ranges of VOC and NOx concentrations.**

### 3.5 Calculation 5. Impact of meteorology on ozone concentrations

It is obvious in Fig. 11 that at any given value of VOC and NOx concentrations, ozone concentration increases as the season

progresses. Although our primary focus is seasonal trends in the photochemical regime, we have also performed box model

calculations to analyze the trend in the ozone concentration itself. We have calculated temperature, absolute humidity and noon-

time solar zenith angle sensitivities, $S_T$, $S_{AH}$, $S_\theta$, for the 24 models, obtaining the approximate ranges given in column 2 of Table

5. The third column summarizes the change in the variable over the course of the season and was extracted from Equation 1 and



325 Fig. 4. Column 4 gives the percentage change in the variable through the season. Multiplying columns 2 and 4 yields column 5, an estimate of the percentage change in ozone concentration induced by the change in each variable. All three variables exert a positive influence on ozone concentration. The predicted total change in ozone concentration is in the range of 40% to 65% from December to March. Contributions from temperature and absolute humidity are much weaker than that from the solar zenith angle. We conclude that most of the increase in ozone concentration observed over the course of the winter at constant $NO_x$ and VOC

330 concentration is driven by the change in actinic flux.

Table 5. Sensitivities of ozone concentrations to noon-time solar zenith angle ($\theta$), temperature (T), and absolute humidity (AH).

| $x =$ | Range of $S_x$ (25th to 75th percentile) | Range of $x$ | % change in $x$ from December to March | % change in $[O_3]$ from December to March |
|---|---|---|---|---|
| $\theta$ | −1.1 to −1.6 | 64° to 42° | −33% | 36% to 53% |
| T | 0.25 to 0.65 | 263 to 279 K | 16% | 4% to 10% |
| AH | 0.02 to 0.04 | 2.7 to 4.2 mbar | 56% | 1% to 2% |


Near the solstice, the solar zenith angle is insensitive to the date, so to calculate $S_0$, we modulated the latitude instead. The large value of $S_0$ helps explain the rarity of the winter ozone phenomenon. Apparently, it is only expected in a narrow range of latitudes. The Uinta and Upper Green River Basins are respectively at about the 40th and 42nd parallels. At just the 45th parallel, with a 12% increase in noon-time solar zenith angle relative to the Uinta Basin, we can expect ozone concentrations to decrease by about 13%

340 to 19%. Therefore, we expect the oil and gas fields of Alberta and Alaska to be spared from winter ozone. Since snow cover is also required for winter ozone, we also expect it to be rare at lower latitudes (Mansfield and Hall; 2018). If global warming causes the snow line to drift farther north, high winter ozone concentrations may become rare even at the 40th to the 42nd parallel.


## 6 Discussion and conclusions

The trend in photochemical indicators documented in Figs. 2 and 3 is dominated more by an increase in $NO_x$ sensitivity than a decrease in VOC sensitivity. $S_{NOx}$ increases from values below $S_{VOC}$ and near zero while $S_{VOC}$ remains relatively flat. In the early

350 season, $S_{NOx}$ is often negative.

Figure 11 demonstrates that two separate effects are responsible for the trend in photochemical indicators documented in Figs. 2 and 3:

355 First, meteorological drivers dominated by the solar zenith angle push the indicator curves to higher levels as the season progresses, increasing and decreasing, respectively, the extents of the $NO_x$-sensitive and $NO_x$-saturation domains. Temperature and absolute humidity also drive the indicator curves upward, but their impact is smaller.





Second, the downward trend in $NO_x$ concentration documented in Fig. 5 pushes typical concentration ranges, the white rectangles

in Fig. 11, downward. In late December, the rectangle lies predominantly in the $NO_x$-saturation domain while in early March it

lies in the $NO_x$-sensitivity domain. This downward trend in $NO_x$ concentration is probably the result of both dilution and emission

effects. Dilution occurs because the typical mixing height increases with the passing season. Emissions change because there are

processes and equipment linked to the temperature. Methane concentrations (and presumably by extension, non-methane VOC

concentrations) also decrease with the advancing season.


The results in Fig. 11 indicate that, all else being equal, ozone concentrations intensify as the season progresses. The calculation

in Table 5 indicates that the solar zenith angle is the most important variable driving this increase.

On the basis of these results, we recommend that Uinta Basin ozone mitigation be focused on controlling both $NO_x$ and VOC. $NO_x$

controls in early winter could conceivably stimulate higher ozone (whenever $S_{NO_x} < 0$), but there are fewer daily exceedances then

(Fig. 12) with lower ozone on average (Fig. 11), and any early-winter ozone increases will probably be more than offset by

decreases in February and March.

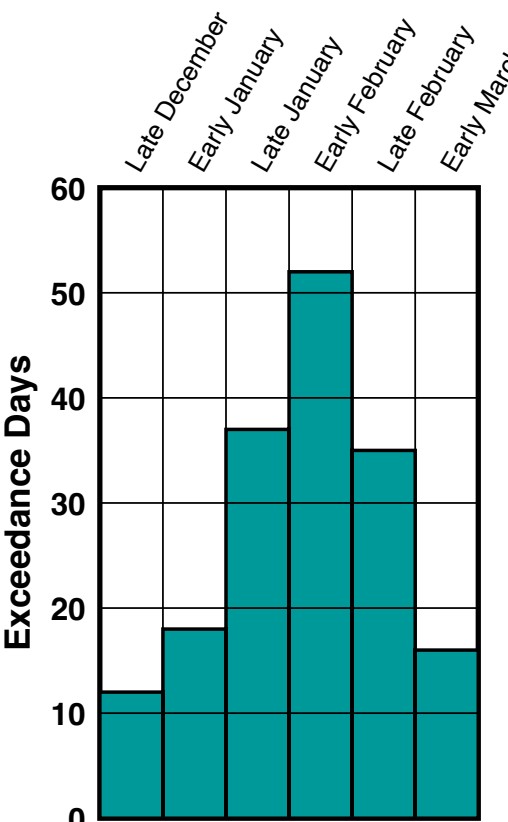

**Figure 12. The number of days in each of the six fortnights between December 2009 and March 2020 during which the 8-hour average ozone concentration at the Ouray monitoring station (central Uinta Basin) exceeded 70 ppb.**



As already mentioned, seasonal trends in the photochemical regime are very common throughout the Northern Hemisphere (Kleinman, 1991; Jacob et al., 1995; Liang et al., 1998; Martin et al., 2004; Jin et al., 2017). This study identifies the primary

drivers for the trend in the Uinta Basin from late December to early March. No doubt these drivers have a similar effect elsewhere, but we should be cautious in extending these results to other regions. For example, biogenic emissions are probably a more important driver of seasonal trends in many regions than they are in the arid Uinta Basin. However, the fact that such trends are ubiquitous may derive from the fact that the actinic flux is the single most important driver in all regions.

**Acknowledgements**

Dr. Liji M. David, of Ramboll, performed the analysis of the OMI satellite data.

Funding for this work was provided by the Utah State Legislature and Uintah Special Service District 1, Uintah County, Utah.

**Competing Interests**

The authors have no competing interests.

**Author Contributions**

SNL compiled the observational data and developed the initial F0AM code. MLM ran the F0AM models, analysed their results, and composed the manuscript.

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
