# Peer review of "Seasonal trends in the wintertime photochemical regime of the Uinta Basin, Utah, USA"

_EGUsphere, 2024_

## Author Response (AR1)

Response to reviewers

At times, the reviewers have requested clarifications that could best be answered by performing additional model runs, but we believe such requests are beyond the scope of the study. Moreover, because of personnel changes at our Institute, running such models would place an undue burden on our staff and cannot be accomplished.

**Reviewer 1**

1. He/she requests a more detailed explanation of the approach used to constrain VOC concentrations. Two sentences beginning with "The VOC speciation profile … ," line 197, have been added. The reviewer's additional request, to examine sensitivity to variations in VOC speciation, would require additional modeling, and we feel this is beyond the scope of the current study.

2. He/she requests additional discussion of the chemistry. In response, we have added a paragraph to the introduction, Lines 33 - 44.

**Reviewer 2**

1. The line about summertime NOx sensitivity has been struck from the abstract.

2. The modifier "multi-day" has been added at line 24.

3. Definitions of Ox and NOz are now supplied at line 47.

4. We are using more restrictive definitions of these terms. So we will keep the word "or," but we have also added the parenthetical phrase "(in the more restrictive …)" at line 65.

5. We chose to use our own VOC speciation data since it was more recent and more extensive. We have added the phrase "based on a more recent, extensive measurement set …" at line 75.

6. A new section, Section 2.1, lines 79-87, has been added to describe the measurement protocols. No filtering of data to improve the correlation between Ox and NOz was applied.

7, 8. It is true that we only used data from the Horsepool station. Emission sources do vary throughout the Basin (e.g., population centers versus isolated areas, different ratios of oil versus gas production). So it is conceivable that the photochemical regime may be different in those areas. However, addressing seasonal changes in photochemical regime at other locations in the Basin would require additional modeling and is beyond the scope of this manuscript. As it is, it was a major undertaking for us to generate the 24 isopleth plots presented here. We have high confidence the same trend (i.e., moving towards more NOx sensitivity later in the winter) occurs throughout the Basin, for reasons described in the text. Because of the size of the Basin relative to the resolution of the OMI instrument, we are not confident the suggested

method would be effective.  While we agree that a study of other Uinta Basin locations would be useful, we believe that the points made in the paper are well supported using just the Horsepool location.

9.  Figures 2 and 3 have been modified as suggested.

10.  The selection of sites is explained completely in the two Mansfield & Hall papers cited at line 140.  We don't feel there is a need to add to the discussion here.

11.  The modifier "multi-day" has been inserted at line 149.

12.  The two figures have been moved to the Supplemental Information.

13.  We see no need for any revision here.  After all, we did say "may behave similarly" at line 181.

14.  We have deleted those sentences.

15.  We addressed this question in paragraph 5 above.

16.  As explained in Section 2.5, we are using "S" to represent a generic sensitivity.  We only add a subscript when necessary to indicate sensitivity to something.

17.  We assume confusion has arisen because we said we analyzed 24 high-ozone "days."  We have reworded the first paragraph of section 3.1 and Table 3 to indicate that we selected 24 multi-day inversion episodes.  That includes a significant majority of the ozone events during the indicated period.  Figure 10 shows us that there were 160 exceedance days total.  If each episode is 5 days long on average, that only gives 32 episodes.  24 episodes do not constitute the complete list, but they come close.

18.  We modified the sentence to read "agreed with the peak-ozone measurement," line 237.

19.  We clearly explain at line 261 that we are probing the effect of changing just one variable.  And frankly, it's just a different way, in the words of the reviewer, to "scale a variable within a certain range."  We applied a different, yet still typical, value of that one variable.

20.  We disagree and choose to keep the figure as is.  The slope of the line tells the reader whether the variable drives an increase or a decrease in sensitivity.  (See also paragraph 21 below.)

21.  We feel that Figure 8 y-axis labels are adequately descriptive.

22.  Perhaps, but we believe the Table is useful and we prefer to keep it.  It indicates at a glance the important driving variables.

23.  We believe that modifying the diagrams to indicate which pixels were calculated and which were krigged would make them extremely difficult to read.  To clarify the question, we have indicated in Section 3.3 the resolution (one-tenth) at which pixels were calculated.  And let me finish with a positive plug for krigging.  It is an excellent two-dimensional interpolation procedure, especially at 10% resolution and with all pixels calculated at the boundaries.  Those pixels may be estimated, but they are good estimates.

**Dr. Tonnesen**

1. Apologies to Dr. Tonnesen for missing her reference during the first go-round.  It has now been added to the manuscript.

2.  Figure 3 does include satellite data for all days in the indicated periods.  We agree that limiting the analysis to high ozone days would be useful, but it would also intensify the concerns you mentioned about a shallow boundary layer.  Figures 2 and 3 are not the central crux of the paper – they only indicate trends that we evaluate exhaustively in the remainder of the text.  The possibility of bias in column data when we are only interested in a shallow boundary layer is a significant, open question.  And one that is beyond the scope of the current work.

3.  All the axis labels have been converted to ppb or ppm units.  For reasons mentioned above, we cannot do any runs at higher NOx.

---

## Author Response (AR2)

We made the following changes to satisfy the requests of the reviewer:

The sentence beginning at line 40, "In the opposite case …"  has been reworded.

Two sentences beginning line 199, "Obviously, use of the same organic speciation profile …" and "Nevertheless, since the speciation was derived …" have been added.

Moreover, coloration has been removed from all tables.